



# A predictive equation for wave setup using genetic programming

Charline Dalinghaus[1], Giovanni Coco[1], and Pablo Higuera[2,3]

[1]School of Environment, Faculty of Science, The University of Auckland, New Zealand
[2]Department of Civil and Environmental Engineering, The University of Auckland, New Zealand
[3]Department of Civil and Environmental Engineering, National University of Singapore, Singapore

**Correspondence:** Charline Dalinghaus (charline.dalinghaus@auckland.ac.nz)

**Abstract.** We applied machine learning to improve the accuracy of present predictors of wave setup. Namely, we used an evolutionary-based genetic programming model and a previously published dataset, which includes various beach and wave conditions. Here, we present two new wave setup predictors, a simple predictor, which is a function of wave height, wavelength, and beach slope, and a fitter, but more complex predictor, which is also a function of sediment diameter. The results show that
the new predictors outperform existing formulas. Therefore, we conclude that machine learning models are capable of not only improving prediction capability (when compared to classical predictors) but also of providing physically sound descriptions of the processes modelled.

## 1 Introduction

As the climate changes, coastal flooding is predicted to increase worldwide. Among the processes included to determine coastal
flooding, wave runup is recognized as one of its major contributors. Defined as the maximum vertical excursion of water above the mean water level, wave runup represents the action of the waves on the beachface. It comprises two different processes: wave setup and swash. Its importance can be highlighted by the fact that neglecting the wave contribution to coastal flooding can result in up to a $\sim 60\%$ underestimation of the flooded area (Vousdoukas et al., 2016).

Wave setup (hereafter referred to simply as setup) is defined as the time-averaged additional elevation of the water level due
to breaking waves (Longuet-Higgins and Stewart, 1964). According to the same authors, as waves approach the shoreline, their action induces the cross-shore transport of momentum, producing changes in pressure and velocity. To conserve the flow of momentum when meeting obstacles, like a sloping beach, it is necessary to account for the action of a force known as radiation stress. This force is proportional to the wave energy and can be written as follows:

$$S_{xx} = E\left(\frac{2kh}{\sinh 2kh} + \frac{1}{2}\right) \tag{1}$$

where $S_{xx}$ is the flux of momentum in the direction of wave propagation, $k = 2\pi/L$ is the wavenumber, $L$ is the wavelength, and $h$ is the still water depth. $E$ is the wave energy per unit surface area, defined as $E = \frac{1}{8}\rho g H^2$, where $\rho$ is the density of water, $g$ is the gravitational acceleration, and $H$ is the wave height. Inside the surf zone, and assuming shallow water conditions, the radiation stress expression can be simplified to:

$$S_{xx} = \frac{3}{2}E = \frac{3}{16}\rho g H^2 \tag{2}$$





Variations in radiation stress result in a rise (setup) and fall (set-down) in the mean water level respectively shoreward and seaward the waves' breaking point (Bowen et al., 1968). Besides being an important component of coastal flooding (Vitousek et al., 2017; Melet et al., 2020) directly impacting the design of coastal structures, setup is also important to the nearshore circulation, such as undertow currents and groundwater flows (Longuet-Higgins, 1983). Ultimately, setup is an important component in the flow circulation and so to sediment exchanges between the sub-aerial and submerged beachface. Thus,

understanding and being able to predict wave setup is vital to protect coastal resources and people living near the shore in a more effective way.

The setup contribution to extreme water levels was first noticed in 1938 during a hurricane on the east coast of the USA, where a water level 1 m higher than in calm water conditions was observed on an exposed beach (Saville, 1961). After this event, many laboratory experiments and field measurements have been conducted using Eq. (2) as the initial point to predict

setup across the surf zone (Bowen et al., 1968; Battjes, 1974; Guza and Thornton, 1981; Holman and Sallenger Jr, 1985; King et al., 1990; Yanagishima and Katoh, 1990; Hanslow and Nielsen, 1993; Raubenheimer et al., 2001; Stockdon et al., 2006; Ji et al., 2018; O'Grady et al., 2019). As a result, empirical setup predictors based on wave parameters, beach morphology, and surf zone processes have been established (Dean and Walton, 2009; Gomes da Silva et al., 2020). Some of the most relevant will be presented next.

Bowen et al. (1968) performed a laboratory investigation of monochromatic waves and related the setup gradient to the beach slope ($\beta_s$) and the ratio of wave height to the mean water depth ($\gamma$) as:

$$\frac{d\bar{\eta}}{dx} = \frac{\beta s}{1 + 8/3\gamma^2} \tag{3}$$

where $\bar{\eta}$ is the setup inside the surf zone, $x$ is the cross-shore coordinate, and $\gamma = H/(\bar{\eta} + h)$ assumes that the ratio between the height of a broken wave or bore ($H$) and the water depth ($h$) remains approximately constant. Their results indicated that

the theory underpredicts the measured setup values, especially at the shoreline, where the maximum setup occurs.

Battjes (1974) performed laboratory experiments and, using Eqs. 2 and 3, estimated the maximum setup ($\bar{\eta}_M$) at the shoreline as:

$$\bar{\eta}_M = 0.38\gamma H_b \tag{4}$$

where $H_b$ is the breaking wave height. King et al. (1990), using the same linear function of incident wave height but replacing

$H$ for $H_{rms}$ (*root mean square*), was also able to accurately predict setup for a random wave field. The authors (but also Guza and Thornton, 1981) highlighted the fact that $\gamma$ values in field observations are much lower than in laboratory experiments.

Through field data measured on a gently sloping beach, Guza and Thornton (1981) correlated maximum setup to the offshore significant wave height ($H_{s0}$):

$$\bar{\eta}_M = 0.17 H_{s0} \tag{5}$$

This predictor underestimated setup at the shoreline, further suggesting that the slope of the setup is not constant across the surf zone, as already seen by previous works (Bowen et al., 1968; Battjes, 1974). Later, Holman and Sallenger Jr (1985)




found a more accurate correlation than only using $H_{s0}$ by relating setup and the surf similarity parameter (Iribarren number: $\xi = \beta_s/(H_{s0}/L_0)^{0.5}$), as presented by Guza and Thornton (1981). However, when isolating low tide data, no significant trend was found with $\xi$, indicating the probable setup dependency on the entire surf zone's bathymetry and not only on the foreshore slope. The same linear relationship between setup and offshore wave height, influenced by tidal fluctuations and the local bathymetry, was also found by Raubenheimer et al. (2001).

Considering the difficulty of defining the parameters used in the predictors above for natural beaches (instead of laboratory environments), Stockdon et al. (2006) proposed a simple empirical parameterization for setup. The equation (Eq. (6)) was based on an extensive dataset, 10 experiments from the USA and the Netherlands, comprising a variety of beach characteristics and wave conditions. The predictor proposed was:

$$\bar{\eta}_M = 0.35\beta_f(H_{s0}L_0)^{0.5} \tag{6}$$

where $\beta_f$ is the foreshore slope and $L_0$ the offshore wavelength obtained using the peak period ($T_p$). Stockdon et al. (2006) found setup is best parameterized when considering offshore over onshore wave hydrodynamics and using the foreshore slope instead of the surf zone slope. Moreover, for fully dissipative conditions, the inclusion of $\beta_f$ in the parameterization is not even necessary. The role of deep water waves and the inclusion of foreshore slope at steeper beaches was also previously recognized by Hanslow and Nielsen (1993).

Recently, Ji et al. (2018) proposed an empirical formula for maximum setup based on different beach slopes and wave parameters through the use of a coupled wave-current model over a linear bathymetry. Besides beach slope, their results showed that setup is also related to wave steepness ($H_{s0}/L_0$):

$$\bar{\eta}_M = 0.220(\beta_s)^{0.538}H_{s0}\left(\frac{H_{s0}}{L_0}\right)^{-0.371} \tag{7}$$

Similar results confirming the role of wave height, beach slope, and wave steepness on maximum setups were found by Yanagishima and Katoh (1990) and by O'Grady et al. (2019). O'Grady et al. (2019) tested different empirical equations and identified that deep water wave height explains 30% of setup variance, followed by an improvement of up to 12% if beach slope is added to the relationship and a further 12% when including wave steepness. Presently, among all studies providing empirical predictors of setup, the most widely used formulation is the one from Stockdon et al. (2006).

Despite an approximately linear relationship between setup at the shoreline and wave height, traditional setup estimates usually do not account for all the complex processes involved in the environment, often translating into significant scatter in predictions (Stephens et al., 2011; Stockdon et al., 2006; Gomes da Silva et al., 2020). Additional factors that may affect the accuracy of setup predictors include: possible errors in the measurements (Guza and Thornton, 1981; King et al., 1990; Lentz and Raubenheimer, 1999), misinterpreted average position of the waterline and difficulty in detecting the maximum setup (Guza and Thornton, 1981; Holman and Sallenger Jr, 1985; King et al., 1990; Lentz and Raubenheimer, 1999), simplifications, uncertain or unaccounted for terms such as bottom stress, alongshore bathymetric features and infragravity waves (Lentz and Raubenheimer, 1999; Ji et al., 2018; O'Grady et al., 2019). In an attempt to overcome these problems and reduce scatter,





innovative data-driven approaches, such as machine learning, are becoming increasingly popular since they can provide rapid
and accurate predictions (Goldstein et al., 2019; Beuzen and Splinter, 2020).

Machine Learning (ML) is a field of computer science focused on developing algorithms that discover relationships between variables by self-learning from a given dataset, without being explicitly programmed to solve that particular problem. Over the past few years, published works have explored the range of applicability of ML approaches, resulting in higher performance and more cost-effective predictors (Goldstein et al., 2019). In coastal sciences, some of the most widely used techniques are
$k$-Nearest Neighbors, Decision Trees, Random Forests, Bayesian Networks, Artificial Neural Networks, and Support Vector Machines (Beuzen and Splinter, 2020). Less known, yet powerful, an algorithm that can provide further insights on the impacts of the underlying processes is Genetic Programming (GP). One of the main advantages of this approach is the ability to develop reliable, robust, and reproducible predictors. Moreover, it is proven to be a powerful technique capable not only of improving predicting capability but also of providing physical insights into coastal processes (i.e., being interpretable) (Passarella et al.,
2018). Studies using GP have focused on developing predictors for wave (Karla et al., 2008; Kambekar and Deo, 2012) and wave ripple (Goldstein et al., 2013) characteristics, sea level (Ghorbani et al., 2010), particle settling velocity (Goldstein and Coco, 2014), open-channel flow mean velocity (Tinoco et al., 2015), swash (Passarella et al., 2018), water turbidity (Wang et al., 2021) and runup (Franklin and Torres-Freyermuth, 2022). GP results usually performed better (in terms of minimizing prediction errors) than those from other commonly used algorithms. Overall, machine learning has shown great promise for
modelling coastal processes, and to the authors' knowledge, it has never been applied to predict wave setup. The amount of available data provides a unique opportunity to develop a novel and more accurate predictor.

In this paper, we propose improving the predictability of wave setup using an evolutionary-based genetic programming model. The paper is organized as follows: Section 2 describes the data, model setup, and model evaluation methods. In Sect. 3, we present the model results and the evaluation of the wave setup equation and compare the newly developed empirical
formulae with several other existing formulations. Section 4 discusses the results obtained and limitations of this approach. Finally, we present the conclusions in Sect. 5.

## 2 Methodology

The increase in spatial and temporal extents, higher resolution, and faster turnaround from acquisition to availability of data related to coastal systems has open up endless possibilities for data-driven algorithms like genetic programming. In this section
we present the data used in this work and the preprocessing methodology followed (2.1); the evolutionary genetic programming model (2.2); and the methods used to evaluate the model predictions accuracy against the testing data and some of the most widely known predictors in the literature (2.3).

### 2.1 Data

In this work, a dataset representing a large variety of beach and wave conditions compiled by Stockdon et al. (2006) has
been used to develop a predictor of wave setup. The data is freely available, and details on how to access it can be found in


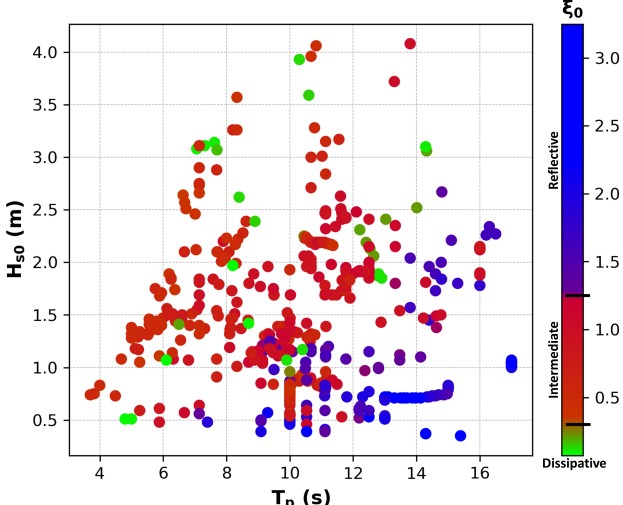

**Figure 1.** Environmental parameters of the dataset: deep-water wave height ($H_{s0}$) versus peak period ($T_p$). The colors represent the Iribarren number ($\xi_0$) where values $<0.3$ (green) characterize dissipative beaches, values $>1.25$ (blue) characterize reflective beaches, and values between both (red) characterize intermediate beaches.

the Code and data availability section. The dataset contains measurements of: maximum setup ($\bar{\eta}_M$), foreshore beach slope ($\beta_f$), median sediment diameter ($D_{50}$), and associated offshore wave characteristics ($H_{s0}$ – significant wave height, and $T_p$ – peak period) from 10 field experiments on sandy beaches resulting in a total of 491 measurements. From these measurements, additional parameters such as the offshore wavelength ($L_0 = gT_p^2/2\pi$) and the Iribarren number ($\xi_0$) were calculated. Table 1

provides full details of the dataset used in this work, including the location and dates of the experiments and the range and average conditions of the environmental parameters. Figure 1 shows the range for some of the parameters available in the dataset. Beach types vary from highly dissipative ($\bar{\xi}_0 = 0.11$ in Terschelling) to fully reflective ($\bar{\xi}_0 = 2.17$ in San Onofre) mean conditions with $\bar{\eta}_M$ ranging from 0.00 (Terschelling) to 1.55 m (Duck 94).

### 2.1.1 Training and testing sets

The target of ML is to use observed data to develop a model able to predict future (unseen) instances. In that sense, the first step is to preprocess the data by normalizing all variables and splitting the dataset into training and testing sets. The training set is used to build and optimize the model, while the testing set is used to quantify the model's performance (i.e., its ability to generalize).

There is no general consensus on which method should be used to split the dataset. In our case, we sought to include the most 135 representative cases from the entire dataset, guaranteeing that the most diverse environmental conditions were well represented in our training set. In addition, the model's aim was not to learn on the largest dataset but to achieve data comprehensiveness





**Table 1.** Range and average (in brackets) environmental conditions for Stockdon's compiled database (Stockdon et al., 2006). Notice that at times only the average value is available.

| Site (Experiment) | Date (Data Points) | $\bar{\eta}_M$ (m) | $H_{s0}$ (m) | $T_p$ (s) | $\beta_f$ | $D_{50}$ (mm) | $L_0$ (m) | $\xi_0$ |
|---|---|---|---|---|---|---|---|---|
| Duck, NC (Duck 82) | 5-25 Oct 1982 (36) | 0.07-1.50 (0.78) | 0.48-4.08 (1.71) | 6.30-16.50 (11.86) | 0.09-0.16 (0.12) | (0.75) | 61.92-424.71 (233.61) | 0.68-2.38 (1.48) |
| Scripps Beach, CA (Uswash) | 26-29 Jun 1989 (41) | 0.06-0.33 (0.18) | 0.54-0.84 (0.69) | (10.00) | 0.03-0.06 (0.04) | (0.20) | (156.00) | 0.41-0.94 (0.58) |
| Duck, NC (Delilah) | 6-19 Oct 1990 (138) | 0.11-1.01 (0.49) | 0.52-2.51 (1.40) | 4.68-14.79 (9.25) | 0.03-0.14 (0.09) | (0.36) | 34.17-341.24 (139.58) | 0.40-1.77 (0.91) |
| San Onofre, CA | 16-20 Oct 1993 (59) | 0.23-0.81 (0.50) | 0.51-1.07 (0.81) | 13.00-17.00 (14.87) | 0.07-0.13 (0.10) | (0.20) | 263.64-450.84 (348.81) | 1.51-2.72 (2.17) |
| Gleneden, OR | 26-28 Feb 1994 (42) | 0.30-0.87 (0.64) | 1.83-2.25 (2.06) | 10.45-16.00 (12.36) | 0.03-0.11 (0.08) | (0.40) | 170.36-399.36 (244.38) | 0.26-1.23 (0.86) |
| Terschelling, NL | 2-22 Apr 1994 (6) | 0.05-0.51 (0.27) | 1.41-3.93 (2.84) | 6.50-10.60 (8.73) | 0.02-0.03 (0.02) | (0.22) | 65.91-175.28 (122.14) | 0.13-0.21 (0.15) |
| Terschelling, NL | 1-21 Oct 1994 (8) | 0-0.10 (0.05) | 0.51-1.97 (1.09) | 4.80-10.40 (7.89) | 0.01-0.02 (0.01) | (0.22) | 35.94-168.73 (104.20) | 0.07-0.25 (0.11) |
| Duck, NC (Duck 94) | 3-21 Oct 1994 (52) | 0.27-1.55 (0.80) | 0.73-4.06 (1.89) | 3.82-14.77 (10.51) | 0.06-0.10 (0.08) | 0.25-2.00 (0.65) | 22.76-340.32 (182.66) | 0.36-1.39 (0.82) |
| Agate Beach, OR | 11-17 Feb 1996 (14) | 0.20-0.65 (0.38) | 1.85-3.14 (2.48) | 7.06-14.32 (11.85) | 0.01-0.02 (0.02) | (0.20) | 77.76-319.90 (228.53) | 0.10-0.22 (0.16) |
| Duck, NC (SandyDuck) | 3-30 Oct 1997 (95) | 0.01-0.91 (0.32) | 0.35-3.57 (1.37) | 3.70-15.39 (9.48) | 0.05-0.14 (0.09) | 0.90-1.65 (1.13) | 21.36-369.49 (151.66) | 0.32-3.25 (1.12) |

with a rather small sample, to later benchmark the model performance against a larger test set. Hence, for this work, we chose the maximum dissimilarity algorithm (MDA) (Camus et al., 2011) as the selection routine.

The MDA aims to select points within the series that are the most dissimilar, ensuring the environment's most diverse representation from the original 491 data measurements (Camus et al., 2011). Each data point is a seven dimensional vector consisting of all the variables in the dataset ($\bar{\eta}_M$, $H_{s0}$, $T_p$, $\beta_f$, $D_{50}$, $L_0$, and $\xi_0$). During the selection phase, the parameters are normalized between 0 and 1 to receive the same weight in the similarity criteria. The calculation starts by selecting an extreme case. We used the largest wave setup value ($\bar{\eta}_M$ = 1.55 m) and its related variables as the initial data point. Subsequent points are picked based on the maximum dissimilarity (i.e., largest distance) with respect to the previously selected cases, which no

longer participate in the selection. The selection of cases ends when the algorithm reaches the number of points determined by the user, after which the denormalized (i.e., original data) training and test sets are reported.





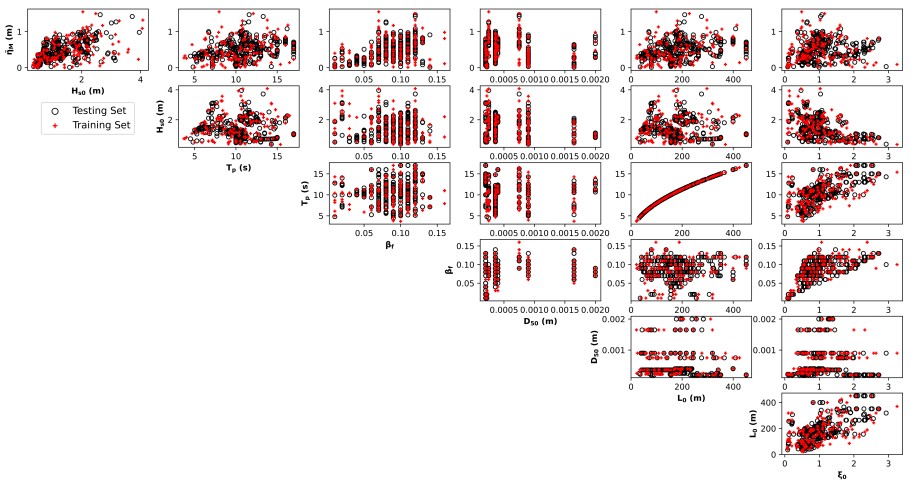

**Figure 2.** Results of MDA selection and correlation between the variables of the dataset. In red is the training set ($\sim 30\%$ of the original dataset), and in black is the testing set ($\sim 70\%$). Environmental variables considered: maximum wave setup ($\bar{\eta}_M$), deep-water significant wave height ($H_{s0}$), peak period ($T_p$), foreshore beach slope ($\beta_f$), median sediment diameter ($D_{50}$), deep-water wavelength ($L_0$), and Iribarren number ($\xi_0$).

MDA was applied to select 150 data points ($\sim 30\%$ of the original dataset), which form the training set. The remaining data ($\sim 70\%$) was used as the testing set to evaluate the model's ability to generalize. Besides, having a more extensive testing set ensures a more accurate estimation of model performance, and by avoiding a large training set, we prevent overfitting. The results of MDA selection are shown in Fig. 2.

## 2.2 Genetic Programming

Genetic Programming (GP) is an evolutionary computational method for which computers automatically solve a problem without requiring a functional prediction form in advance (Koza and Poli, 2005; Poli et al., 2008). In GP, individuals of a population are computer programs (i.e., equations) of varying size and shape that genetically "breed" (Koza, 1992). The separate elements forming the equations (variables and mathematical operators) represent each individual's chromosomes. Inspired by natural selection and the "survival of the fittest", GP uses an initial population of equations where the fitter ones (parents) are selected to breed a new generation of offspring (i.e., new equations). At each generation, a new population is created through the application of genetic operations (evolutionary process): reproduction, crossover and mutation. In the end, the final optimized predictor (within user-defined expression complexity limits) can be represented in mathematical form. The step-by-step process involved in implementing the GP model is illustrated in Fig. 3 and further explained as follows:

[1] Initialization. An initial population of random equations is created by selecting a set of independent variables, mathematical operators, and constant values, which are introduced in agreement with the control parameters of the model set by the user (see Table 2). It is important to highlight that GP does not require non-dimensional (normalized) inputs.




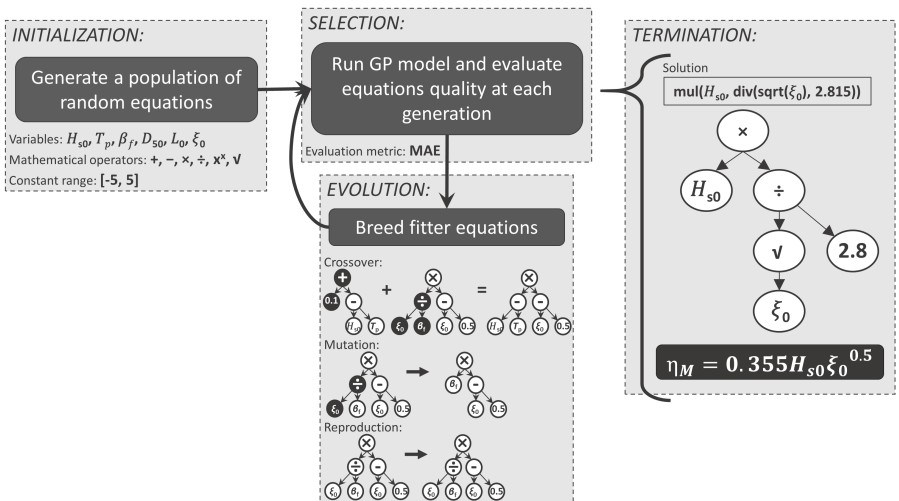

**Figure 3.** Main loop of the GP model (inspired by Poli et al. (2008)). The equations encoded as a tree with variables, operators, and coefficients shown in the evolution framework are examples of genetic operations for the reader's easy visualization. The solution shown in the termination framework is the same one presented later in the Results section.

[2] Selection. "Tournament" selections between equations are realized in order to decide which equations will evolve in the next generation. Among the selected equations for each tournament, chosen at random from the population, the GP model finds the one that best fits the training data (i.e., lowest fitness function). As the fitness function, we selected the mean absolute error (MAE), which is formulated as follows:

$$\text{MAE} = \frac{1}{n} \sum_{i=1}^{n} |M_i - P_i| \tag{8}$$

where $n$ is the number of measured values; and $M_i$ and $P_i$ denote measured and predicted values, respectively.

[3] Evolution. From the best solution, a new solution is created through an evolutionary process. Genetic operations are applied to the winner of each tournament at random. Therefore, fitter individuals are more likely to produce new equations than inferior individuals. New equations for the next generation are created by: (a)Crossover: Merging random chromosomes/parts from two tournament winners; (b)Mutation: Selecting random chromosomes/parts of the tournament winner to change, and; (c)Reproduction: Copy of the tournament winner. A parsimony coefficient is used to penalize large equations, avoiding bloat (larger equations with no significant improvement in fitness).

[4] Termination. The execution of the model stops when the termination criteria is reached. The final solution (i.e., equation – encoded as a tree with variables, operators, and coefficients) is the one that reaches the established minimal error (stopping criteria) or the best one at the specific number of generations predetermined by the user.



**Table 2.** Hyperparameters setup of the GPLearn model.

| Parameter | Value |
|---|---|
| Independent Variables | $H_{s0}$ (m), $T_p$ (s), $\beta_f$, $D_{50}$ (m), $L_0$ (m), $\xi_0$ |
| Mathematical Operators | $+, -, \times, \div, x^x, \sqrt{}$ |
| Constant Range | [-5, 5] |
| Population Size | 5000 |
| Generations | 1000 |
| Tournament Size | 20 |
| Fitness Function | Mean Absolute Error (MAE) |
| Genetic Operations | Crossover (70%), Mutation (25%) and Reproduction (5%) |
| Parsimony Coefficient | 0.0005 |
| Stopping Criteria | 0.01 |

The choice of the values above was driven by extensive testing and sensitivity analyses performed as part of this work.

In this work, the GP model was built using the GPLearn Python module (Stephens, 2015), a machine learning library
180  extended from Scikit-Learn (Pedregosa et al., 2011). We have run the model with different setups such as different population sizes (2,000 - 500,000), generations (20 - 10,000), tournament sizes (10 - 1,000), parsimony coefficients (0.001 - 0.01) and genetic operations proportions. Although it is not strictly necessary, we have also run the model using a normalized input. All the runs stopped by reaching the number of chosen generations since we set a very low stopping criteria error. In the end, the best predictor was found through Table 2 model setup, and a minimal or no improvement was achieved with more complicated
185  equations. To select the best predictor, we focused on finding the balance between achieving a low error reduction and high predicting capabilities, and obtaining simpler, physically meaningful equations. The code is available, and details on how to access it can be found in Code and data availability.

### 2.3  Model Evaluation

The testing dataset was used to evaluate the GP predictor's performance through several statistical parameters, including the
190  square of the Correlation Coefficient (the square of Pearson's Correlation - $r^2$ - Eq. (9)), Coefficient of Determination ($R^2$ - Eq. (10)), modified Index of Agreement ($d1$ - Eq. (11)), Mean Absolute Error (MAE - see Eq. (8)), and Root Mean Square Error (RMSE - Eq. (12)).

$$r^2 = \frac{\left(\sum_{i=1}^{n} (M_i - \bar{M})(P_i - \bar{P})\right)^2}{\sum_{i=1}^{n} (M_i - \bar{M})^2 \sum_{i=1}^{n} (P_i - \bar{P})^2} \tag{9}$$

$$R^2 = 1 - \frac{\sum_{i=1}^{n} (M_i - P_i)^2}{\sum_{i=1}^{n} (M_i - \bar{M})^2} \tag{10}$$




$$d_1 = 1 - \frac{\sum_{i=1}^{n} |M_i - P_i|}{\sum_{i=1}^{n} (|M_i - \bar{M}| + |P_i - \bar{M}|)} \tag{11}$$

$$\text{RMSE} = \sqrt{\frac{1}{n} \sum_{i=1}^{n} (M_i - P_i)^2} \tag{12}$$

where $\bar{M}$ and $\bar{P}$ are the corresponding average values of measured and predicted parameters, respectively.

The values of $r^2$ and $R^2$ are measures of linear correlation, where $r^2$ explain the proportion of variance between two sets of data, and $R^2$ is used to evaluate how well the model predicts in comparison to actual measurements (the model's performance). Alternatively, $d_1$ is also used to evaluate the agreement between predicted and measured values. For further details about $d_1$ the reader is referred to Willmott (1981) and Willmott et al. (1985). $r^2$, $R^2$, and $d_1$ are dimensionless, and values closer to 1 represent better agreements. In contrast, MAE and RMSE measure the errors given by the difference between predicted and measured values; in addition, the second penalizes large errors (bad predictions). Both MAE and RMSE are expressed in the same units of $\bar{\eta}_M$ (m), which means that lower values (closer to 0) indicate more accurate predictions.

Because each metric has its own strengths and limitations, the combination of these five different criteria allowed for a more comprehensive comparison between the model results. Moreover, these same statistical parameters were used to compare the present model with other existing predictors, namely the widely used Stockdon et al. (2006) and Guza and Thornton (1981), Holman and Sallenger Jr (1985), Yanagishima and Katoh (1990), Hanslow and Nielsen (1993), Ji et al. (2018) and O'Grady et al. (2019).

## 3 Results

From the multiple equations obtained as an output from the GP model, we selected two predictors of wave setup. A simple predictor is presented in Eq. (13). Alternatively, a more complex but also more accurate predictor which maintains physical interpretability is presented in Eq. (14).

$$\bar{\eta}_M = 0.355 H_{s0} {\xi_0}^{0.5} \tag{13}$$

$$\bar{\eta}_M = \frac{H_{s0}}{4.08} \left( \frac{\xi_0}{3.25} + \frac{\xi_0}{\xi_0 + 0.64} + \frac{\xi_0}{1625 D_{50} + \xi_0} \right) \tag{14}$$

Here, it is important to highlight that the coefficient 1625 in Eq. (14) is dimensional, with units of $m^{-1}$.

Equation (13) stands out for its simplicity. This equation is also very similar to previous predictors found in the literature (e.g. Holman and Sallenger Jr, 1985; Stockdon et al., 2006; Ji et al., 2018; O'Grady et al., 2019). However, the new models presented here differ from previous equations (except Holman and Sallenger Jr, 1985) by considering wave height, wavelength, and


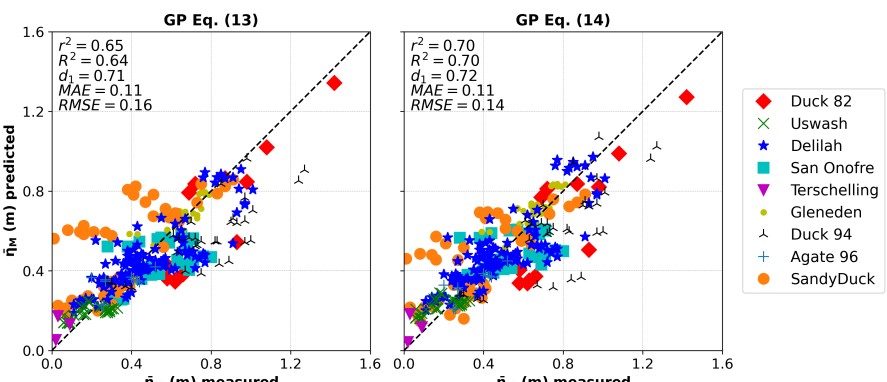

**Figure 4.** Measured versus predicted maximum wave setup ($\bar{\eta}_M$) using the testing data for Eq. (13) (left panel) and Eq. (14) (right panel). The metrics used to evaluate the GP predictors' performance are also presented. $r^2$ = the square of the Correlation Coefficient, $R^2$ = Coefficient of Determination, $d1$ = modified Index of Agreement, $MAE$ = Mean Absolute Error, and $RMSE$ = Root Mean Square Error. Different markers/colors refer to different field experiments, as referenced in the legend.

foreshore beach slope combined, in terms of the non-dimensional Iribarren number. Furthermore, the complexity of Eq. (14) is slightly larger because of the additional terms it includes. Similarly to Eq. (13), the first two terms depend only on the wave height and Iribarren number (i.e., they are informed by wave dynamics and beach slope). The third term in Eq. (14) includes $D_{50}$, measured in m. Therefore, it requires a dimensional coefficient for the predictor to be dimensionally consistent. As a result, this term contains information on the wave dynamics and beach slope, but also on the sediment size. We remark that

this is the first time that grain size is introduced in a wave setup equation.

     Figure 4 presents the scatter plots between measured and predicted $\bar{\eta}_M$ obtained from Eqs. (13) and (14). The data shown in the figures are the testing data and the metrics used to evaluate the GP predictors' performance. Although more complex, Eq. (14) represents the best equation in terms of the lowest error when considering the $RMSE = 0.14$ m, in comparison with a $RMSE = 0.16$ m from Eq. (13) (a 12.5% difference). Nevertheless, note that both equations have the same $MAE = 0.11$ m.

Equation (14) also yields higher values of $r^2 = 0.70$, $R^2 = 0.70$, and $d_1 = 0.72$ as compared to Eq. (13) ($r^2 = 0.65$, $R^2 = 0.64$, and $d_1 = 0.71$), indicating a better fit of the Eq. (14) with the testing data. Furthermore, Eq. (13) and Eq. (14) performed well on beaches with dissipative (Agate 96 and Terschelling) and reflective (San Onofre) conditions. Among all field experiments, Duck 94 (intermediate to reflective with large wave conditions) was the beach that showed less correlation with our models.

     A successful predictor should present physical interpretability, but it also must be coherent with the real environment.

Therefore, Fig. 5 presents a sensitivity analysis with data in the range measured (in red) and outside (extrapolated, in black) to evaluate the influence of the input variables on $\bar{\eta}_M$. In both models we observe a positive correlation between $\bar{\eta}_M$ and $H_{s0}$, and between $\bar{\eta}_M$ and $\xi_0$. As expected, the linear relationship between $\bar{\eta}_M$ and $H_{s0}$ means that larger waves produce greater setups. Regarding Iribarren number, $\bar{\eta}_M$ is proportional to the square root of $\xi_0$ in Eq. (13), and a similar non-linear relationship appears in Eq. (14). In this case, greater setups are likely to occur for reflective beach conditions (higher $\xi_0$). However, the

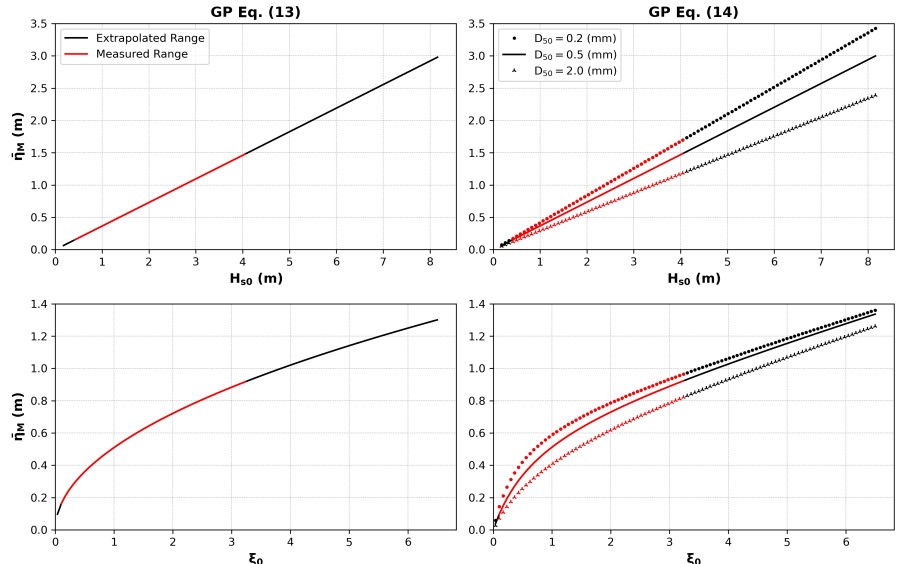

**Figure 5.** General behavior of the maximum wave setup ($\bar{\eta}_M$) predictors presented in Eq. (13) (left panels) and Eq. (14) (right panels), as a function of deep-water significant wave height ($H_{s0}$), Iribarren number ($\xi_0$) and median sediment diameter ($D_{50}$). $D_{50}$ is represented by its minimum (0.2 mm), mean (0.5 mm) and maximum (2.0 mm) values in the dataset. Data within the measured range are depicted with red points. Black points represent an extrapolated range for $H_{s0}$ and $\xi_0$. Note the different Y and X-axis ranges for each graph.

rate of increase in setup decreases with higher Iribarren numbers in both cases. On the other hand, $D_{50}$ (which only appears in Eq. (14)) is negatively correlated with $\bar{\eta}_M$, meaning that greater setups are expected to occur on beaches with smaller sediment diameters. The variation in setup with $D_{50}$ appears to be of lower magnitude in comparison with $H_{s0}$ and $\xi_0$. Although, with increasing $H_{s0}$, the sensitivity of $\bar{\eta}_M$ to the median grain size ($D_{50}$) increases. The same is not valid with $\xi_0$. On the contrary, there is a slight decrease in the sensitivity of $\bar{\eta}_M$ as a function of $D_{50}$ with larger $\xi_0$ values. Here, we have expanded the

predictor's use beyond the range of the measurements that comprise the dataset, to test its general behaviour and stability, showing that the predictors work sensibly also beyond the range of the available measurements. Smaller values of $H_{s0}$ and $\xi_0$ never result in negative $\bar{\eta}_M$ values, and the observed trends continue in the unobserved data range.

   Using the entire dataset (training + testing) we also compared the results of Eq. (13) and Eq. (14) with the most widely known predictors in the literature. Table 3 and Fig. 6 show the performance of nine distinct empirical equations, including the ones

presented in this work, in determining maximum wave setup. Eqs. (13) and (14) show a good agreement with the measured dataset, with less scatter (especially Eq. (14)) if compared with the others. Overall, our ML-driven approach achieved better results with Eq. (14) outperforming all other predictors ($r^2 = 0.64$, $R^2 = 0.64$, $d_1 = 0.70$, MAE =0.12 m and RMSE = 0.17 m). Similarly, Eq. (13) exhibits good results as well ($r^2 = 0.58$, $R^2 = 0.57$, and $d_1 = 0.68$, MAE = 0.13 and RMSE = 0.19), the same ones as Ji et al. (2018)'s equation (although our predictor contains one coefficient less). In comparison, Stockdon

et al. (2006) formulation presents lower metrics results ($r^2 = 0.49$, $R^2 = 0.44$, and $d_1 = 0.66$). Correlation ($r^2 = 0.55$) and





**Table 3.** Statistical metrics of predictors' performance using measured data from Stockdon et al. (2006). We assumed $H_{rms0} = H_{s0}/2^{0.5}$, following Rayleigh distribution in deep water. $\bar{\eta}_M$ = maximum wave setup; $r^2$ = the square of the Correlation Coefficient, $R^2$ = Coefficient of Determination, $d1$ = modified Index of Agreement, MAE = Mean Absolute Error, and RMSE = Root Mean Square Error.

| Author | $\bar{\eta}_M$ **predictor** | $r^2$ | $R^2$ | $d_1$ | MAE (m) | RMSE (m) |
|---|---|---|---|---|---|---|
| Present work - Eq. (13) | $0.355 H_{s0}\xi_0^{0.5}$ | 0.58 | 0.57 | 0.68 | 0.13 | 0.19 |
| Present work - Eq. (14) | $\frac{H_{s0}}{4.08}\left(\frac{\xi_0}{3.25} + \frac{\xi_0}{\xi_0+0.64} + \frac{\xi_0}{1625 D_{50}+\xi_0}\right)$ | 0.64 | 0.64 | 0.70 | 0.12 | 0.17 |
| Guza and Thornton (1981) | $0.17 H_{s0}$ | 0.30 | -0.45 | 0.43 | 0.27 | 0.34 |
| Holman and Sallenger Jr (1985) | $0.46\xi_0 H_{s0}$ | 0.49 | 0.08 | 0.60 | 0.20 | 0.27 |
| Yanagishima and Katoh (1990) | $0.052 H_{s0}\left(\frac{H_{s0}}{L_0}\right)^{-0.2}$ | 0.39 | -0.83 | 0.41 | 0.31 | 0.38 |
| Hanslow and Nielsen (1993) | $0.048 H_{rms0} L_0^{0.5}$ | 0.38 | 0.12 | 0.50 | 0.22 | 0.27 |
| Stockdon et al. (2006) | $0.35\beta_f (H_{s0}L_0)^{0.5}$ | 0.49 | 0.44 | 0.66 | 0.15 | 0.21 |
| Ji et al. (2018) | $0.220\beta_s^{0.538} H_{s0}\left(\frac{H_{s0}}{L_0}\right)^{-0.371}$ | 0.58 | 0.57 | 0.68 | 0.13 | 0.19 |
| O'Grady et al. (2019) | $0.92\beta_f H_{s0}\left(\frac{H_{s0}}{L_0}\right)^{-0.3}$ | 0.55 | 0.51 | 0.68 | 0.14 | 0.20 |

agreement ($d_1 = 0.68$) results showed by O'Grady et al. (2019)'s equation are also close to Eqs. (13) and (7) (Ji et al., 2018). In contrast, their model prediction is worse ($R^2 = 0.51$ as compared to $R^2 = 0.57$). In relation to the error metrics, both O'Grady et al. (2019) and Stockdon et al. (2006) predictors show good results, with MAE = 0.14 and 0.15 m and RMSE = 0.20 and 0.21 m, respectively. Finally, the predictors of Holman and Sallenger Jr (1985) and Hanslow and Nielsen (1993) produce more

scatter and tend to overestimate the results, while Guza and Thornton (1981) and Yanagishima and Katoh (1990)'s predictors largely underestimate the setup values. This results in very low coefficients of determination ($R^2 = 0.08, 0.12, -0.45$, and $-0.83$, respectively), meaning that their predictions match poorly with observations. Here, it is worth pointing out that models with the same correlation coefficient, e.g. Holman and Sallenger Jr (1985) and Stockdon et al. (2006) ($r^2 = 0.49$), present similar patterns. However, the coefficient of determination appears to better describe the accuracy of the model predictions. In this

case, using the coefficient of determination, the Stockdon et al. (2006)'s equation ($R^2 = 0.44$) performs better than Holman and Sallenger Jr (1985)'s ($R^2 = 0.08$).

## 4 Discussion

In this work, we have presented two predictors that demonstrate the predictive capability of genetic programming. The results show that the novel GP predictors outperform existing formulas by presenting fitter equations for the entire dataset compiled

by Stockdon et al. (2006). Additionally, unlike most previous predictors, they also present a good fit for both dissipative (Agate 96 and Terschelling) and reflective (San Onofre) beach conditions. Although the main advantage of the GP model is the possibility of fully exploring multiple equation forms from different model parameters trying to find a more accurate variable

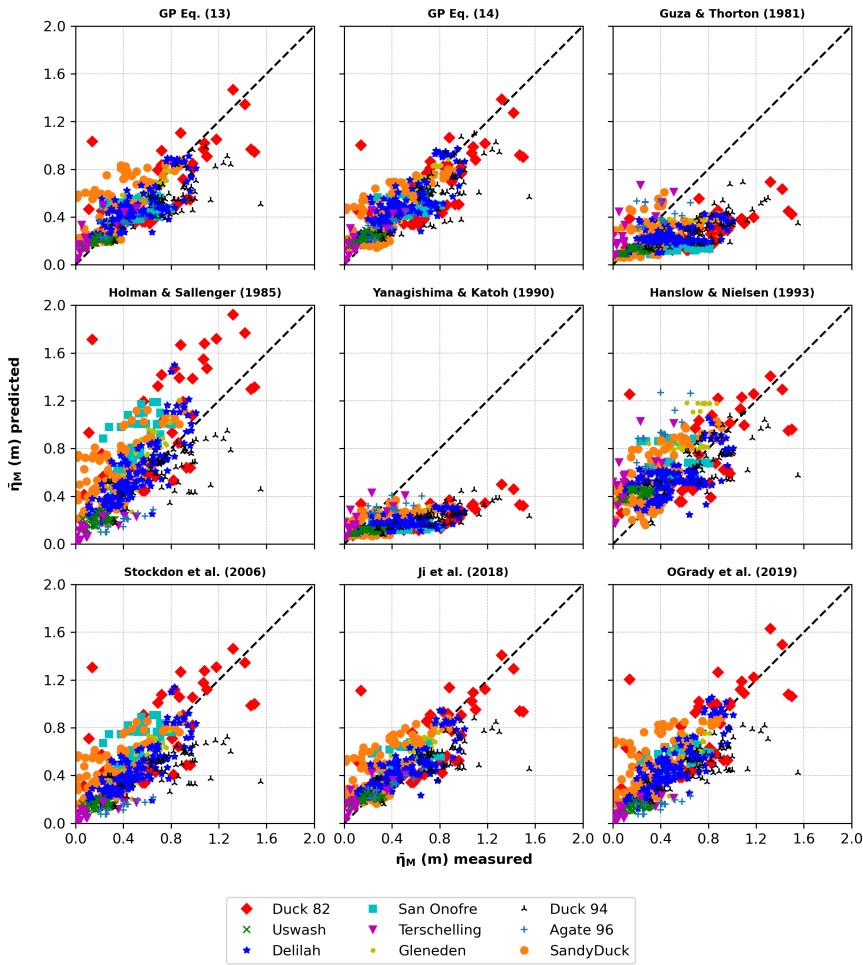

**Figure 6.** Measured versus predicted maximum wave setup ($\bar{\eta}_M$) using the entire dataset (training + testing) for nine distinct empirical equations, including the ones presented in this work. Different markers/colors denote different field experiments.

combination during evolution, the final selection of the proposed solution remains subjective. This last step requires the user to have knowledge on the specific topic, so that the expression chosen is dimensionally and physically correct. Generally, as the

complexity of the solutions increases, the error decreases. Therefore, more complex predictors usually fit the training dataset better than simpler ones. However, they may become too specific for the training dataset, thus, they may lose generalization power (due to overfitting) when applied to different datasets (Tinoco et al., 2015; Passarella et al., 2018). As a result, the proposed solutions should ideally be simple, easy-to-use and to interpret, and have a physical meaning.

The variables ($H_{s0}$, $\beta_f$, $L_0$) presented in the equations are the same used in previous published predictors (Holman and

Sallenger Jr, 1985; Stockdon et al., 2006; Ji et al., 2018; O'Grady et al., 2019) but with a different arrangement. The result is in agreement with Longuet-Higgins and Stewart (1964), who stated the cross-shore gradient of radiation stress is principally





controlled by the wave height. Additionally, in line with O'Grady et al. (2019), we found the wave setup predictor is best parameterized with the inclusion of wave steepness and beach slope (through the Iribarren number) along with the wave height. Although playing a limited role, sediment diameter was also introduced in Eq. (14) improving the performance of the
predictor by establishing a non-linear, inversely proportional relationship with the maximum wave setup.

Most wave setup studies (e.g. Guza and Thornton, 1981; King et al., 1990) present the offshore wave height as the primary contributing factor to wave setup, since it dictates the energy available for the production of setup. Nevertheless, the wave setup is not simply a result of incident waves but also induced by the shape of the beach profile. The recognition of the beach morphology's role ends up improving the predictor (Stephens et al., 2011; O'Grady et al., 2019; Gomes da Silva et al., 2020).
However, there is no consensus on which region to use to estimate the beach slope. Some works include only the foreshore beach slope (Holman and Sallenger Jr, 1985; Hanslow and Nielsen, 1993; Stockdon et al., 2006; O'Grady et al., 2019) into the predictor, and some use the average surf zone beach slope (Bowen et al., 1968; Raubenheimer et al., 2001; Stephens et al., 2011; Gomes da Silva et al., 2020). Despite being complex to quantify, the role of beach slope along with grain size is essential in incorporating the effect of the cross-shore beach profile in estimating wave setup. Although not leading to significant changes
in current predictions, the presence of sediment diameter in Eq. (14) needs careful consideration. As in Poate et al. (2016), who stated the importance of grain size in gravel's beach runup parameterization, here, its addition also improves wave setup prediction. This second order effect could tentatively be related to beach permeability, which increases with sediment size and results in a lower setup. The novel inclusion of $D_{50}$ as a second-order effect may indicate that we still have very limited information to describe an entire beach (e.g., not considering the presence of multiple bar systems). After over 50 years of
research, wave setup prediction still presents a number of issues to be solved to enhance parametric predictors based on environmental variables. It includes the influence of beach permeability (Longuet-Higgins, 1983; Nielsen, 1988, 1989) and tide (Holman and Sallenger Jr, 1985; Raubenheimer et al., 2001; Stockdon et al., 2006) as second order processes subject to discussion.

In the field, different methodologies have been used to measure wave setup. Equipments as resistance wire runup meters
(Guza and Thornton, 1981), manometer tubes (Nielsen, 1988), pressure transducers/sensors (King et al., 1990; Lentz and Raubenheimer, 1999; Raubenheimer et al., 2001), sonar altimeters (Lentz and Raubenheimer, 1999; Raubenheimer et al., 2001), and video cameras (Holman and Sallenger Jr, 1985). O'Grady et al. (2019) suggested that around $\sim 46\%$ of the setup variance is possibly explained by measurement errors or related to critical processes that could not be translated into simple predictors yet, as just highlighted. Since the surf and swash zones are a highly dynamic environment, the bathymetry is rapidly
evolving and changes are difficult to predict. In an attempt to overcome this problem, Ji et al. (2018) used a wave-current numerical model to generate setup data for idealized beach conditions. Although presenting extremely promising results, there is still significant scatter around the wave setup predictor. Accounting for all the effects and processes that may be important for wave setup remains an arduous challenge.





## 5 Conclusions

In this work we proposed two new empirical equations for the maximum wave setup using data compiled by Stockdon et al. (2006) to feed an evolutionary-based genetic programming model. A simple, yet accurate, predictor and a more complex but fitter predictor, which maintains physical interpretability, were tested and evaluated against other seven widely known empirical equations for maximum wave setup. The results of both GP based predictors emphasized similarities with previous ones and incorporated new dependencies. Compared with previous predictors, the new ones demonstrate an improvement in prediction

performance and a goodness of fit for a wide range of environmental conditions, including both dissipative and reflective beaches. The novel predictors are simple, can be easily used in practical applications, and open up new paths for future wave setup research.

So far, only a few studies have addressed wave setup predictions and all past predictors present significant scatter around the data. All predictors share similarities in their structure, possibly indicating that limits in predictability are related to the use

of oversimplified variables, $H_{s0}$, $T_p$, $\beta_f$, and $D_{50}$, that do not fully capture the complexity of surf zone processes. The use of additional parameters (e.g., to better describe the surf zone seabed profile) appears necessary to more accurately describe wave setup in a natural environment.

As additional data become available and better algorithms are developed, more accurate predictors will be generated. Currently, innovative data-driven approaches, such as genetic programming, are able to extract patterns from samples resulting

in higher performance and more cost-effective predictors. Although we still need to deal with data scarcity and measurement uncertainties, our results reveal that the genetic programming model has competence in data generalization and, being a data-driven technique, it will only get more accurate as more data becomes available. Through the use of a data-driven model, we were able to present reliable, robust, and reproducible predictors, able to represent the physical processes behind the available datasets.

Understanding and predicting nearshore processes is vital to protect coastal resources and people living near the shore. We expect that the results of this work will contribute to improving the predictability of wave setup, a key factor in coastal flooding. Additionally, we also seek to stimulate further discussion about the use of machine learning as a powerful data analysis tool and the possibility of its use to improve coastal sciences/management.

*Code and data availability.* The dataset is available in Stockdon and Holman (2011) and also can be downloaded from https://coastalhub.

science/data (Wave Runup Field Data). Implementation of the GP model in Python is publicly available in https://github.com/chardalinghaus/GpLearn_WaveSetup.

*Author contributions.* All authors developed the concept for this study and the methodology. C.D. performed the analysis and wrote the original manuscript. All authors verified the analysis, discussed the results and edited the manuscript.



*Competing interests.* The authors declare that they have no conflict of interest.

*Acknowledgements.* Charline Dalinghaus is supported by a Doctoral Scholarship from The University of Auckland.



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
