# Peer review of "A predictive equation for wave setup using genetic programming"

_Natural Hazards and Earth System Sciences, 2022_

## Author Response (AR1)

**Dear Referee #1 and Dr. Ribas,**

**Thank you for your thoughtful review of our manuscript. We have carefully reviewed your comments and have revised the manuscript accordingly. Please see below our responses to each one of your comments.**

**Referee #1 comments:**

*The paper discusses finding equations for wave setup using machine learning (ML) algorithms, and the main contribution is in applying ML algorithms to the geophysical problems. In general, the paper is clearly presented and well organized. However, I have two main concerns about this paper.*

We thank you for the positive analysis.

*The first question is on the results by ML algorithms. One of the main contributions of this paper is the equation (14), but the physical implication behind it is unclear. As authors understand, complicated equations driven by the ML algorithms will give well-fitting results. At the same time, the equations are meaningful if they are physically interpretable. There are three terms in eq. (14), and the last term reversely relates the setup height (M) with the grain size (D50). In line 297-298, authors mentioned "This second order effect could tentatively be related to beach permeability, which increases with sediment size and results in a lower setup." However, to my knowledge, the permeability is related to the distribution of the grain size, not the average of the grain size.*

We agree that the distribution of the grain size is important for the permeability, but we need to point out that the median grain size is a first-order effect. Previous works such as Krumbein and Monk (1942), Ward (1964), Beard & Weyl (1973) and Sheperd (1989) presented the permeability expressed as a function of median grain size ($D_{50}$). Overall, the results of these studies showed that permeability increases with increasing average particle diameter size. We used this concept, as well as the results presented by Poate et al. (2016) and Power et al. (2019), in a tentative way to explain the physical meaning of $D_{50}$ in our equation. However, please note that we do not claim that the presence of $D_{50}$ in Eq. (14) is, in fact, related to beach permeability, as we do not have how to prove it with this work. Instead, we wanted to raise the discussion of the permeability role in the wave setup again, as previously mentioned by Nielsen (1988, 1989). Please see below:

"The presence of sediment diameter in Eq. (11) also needs careful further consideration. As in Poate et al. (2016) and Power et al. (2019), who stated the importance of grain size in runup parameterization, its inclusion also improves wave setup prediction. This second-order effect could be tentatively related to beach permeability, which increases with sediment size and results in a lower setup. However, the limited amount of sediment diameter data may not be entirely appropriate to claim such finding." (lines 287 – 291 in the new version of the manuscript)

"More importantly, the novel inclusion of $D_{50}$ as a second-order effect may indicate that we still have very limited information to describe an entire beach. Other examples of second-order effects are not considering the presence of multiple bar systems or even incorporating wave direction. After over 50 years of research, wave setup prediction still presents a number of issues to be solved in future works which can enhance parametric predictors based on environmental variables. These also include the influence of beach permeability (Longuet-Higgins, 1983; Nielsen, 1988, 1989) and tide (Holman and Sallenger Jr, 1985; Raubenheimer et al., 2001; Stockdon et al., 2006) as second-order processes

subject to discussion. More recently, works from Guérin et al. (2018) and Martins et al. (2022) investigated the role of the wave-induced nearshore circulation processes (bottom stress, vertical mixing, and vertical and horizontal advection), resulting in an improved wave setup prediction across the surf zone. The contribution of these parameters can be even larger on steeper beach slopes (Martins et al., 2022)." (lines 294 – 302 in the new version of the manuscript)

*The second one is on the sample size and data availability. The sample size of 491 cases is relatively small to apply ML algorithms.*

The ideal sample size for machine learning is a tricky issue. To our knowledge, there is no unique, optimal approach to discovering the ideal sample size; it mainly depends on the complexity of the problem, the distribution of the variables in the data available and the chosen algorithm (genetic programming). In our case, we considered all data with the same acquisition/processing procedure that is freely available. Nevertheless, we looked at previous works using genetic programming to check if a similar amount of data has been used before. Some examples of successful applications using limited datasets include Tinoco et al. (2015), Passarella et al. (2018) and Wang et al. (2021). One point we make in the conclusion is that as more data becomes available, there will be an opportunity to further improve the predictive algorithm (e.g., to reduce the scatter). Please see below:

"As additional data become available and better algorithms are developed, more accurate predictors will be generated. Data-driven approaches are able to extract patterns from samples resulting in higher performance and more cost-effective predictors. Although we still need to deal with data scarcity and measurement uncertainties, our results reveal that the genetic programming model is competent in data generalisation. Being a data-driven technique, it will be more accurate as additional high-quality data becomes available." (lines 329 – 333 in the new version of the manuscript)

*And it seems that more data are available from the provided link (https://coastalhub.science/data). It would be better to mention the reasons to use Stockdon and Holman 2011 data only.*

Indeed, one extra wave setup dataset was available, the one from Gomes da Silva et al. (2018). Through personal communication with the lead author (Gomes da Silva), she indicated that the wave setup data is not reliable (we have already removed this dataset from the https://coastalhub.science/data website). That is why we decided to use Stockdon and Holman's (2011) data only. It was the only freely available dataset with input (physical processes) and output (wave setup) data to train our data-driven model. Please see below:

"To make setup predictions using a data-driven model, it is necessary to have the input and output data to train it. The input data is related to physical processes that induce the output, wave setup. In this work, we used a dataset meeting these requirements, representing a large variety of beach and wave conditions compiled by Stockdon et al. (2006)." (lines 108 - 110 in the new version of the manuscript)

*Moreover, I could not find the grain size (D50) from Stockdon and Holman, 2011 (https://pubs.usgs.gov/ds/602/) or (https://coastalhub.science/data). Authors need to provide a complete data set, and how they acquired the grain size.*

The median grain size data can be found in the 9th column of the wave runup data file at https://coastalhub.science/data. However, these values were obtained from reports and papers describing the beaches: Duck82 – Mason et al (1984); Uswash - Holland et al. (1995); Delilah –

Thornton and Humiston (1996) and http://frf.usace.army.mil/delilah/start; San Onofre - Raubenheimer and Guza (1996); Gleneden – Power et al. (2019); Terschelling - Ruessink et al. (1998); Duck 94 – Stauble and Cialone (1996) and Gallagher et al. (1998); Agate - Ruggiero et al. (2004) and; SandyDuck - www.frf.usace.army.mil/SandyDuck/SandyDuck, rather than from Stockdon and Holman (2011) as mentioned. This is now corrected in the new version of the manuscript as follows:

"Median sediment diameter ($D_{50}$) data was obtained from reports and papers describing the beaches." (lines 116 - 117 in the new version of the manuscript)

*Minor comments*

We thank the Reviewer #1 for the careful reading. The minor comments have been corrected in the updated version of the paper.

*L114 "has open" -> "has opened"* – Removed from the new version of the manuscript.
*L311 "Although presenting extremely promising results" -> ambiguous* – Removed.
*L330 "being a data-driven technique, it will only get more accurate as more data becomes available."*
*Not just more data but high quality data are necessary.* – Corrected (line 333).
*L332 "able to represent"->"representing"* – Removed.

**Dr. Ribas comments:**

*This article presents a new application of genetic algorithms to obtain empirical formulas for the maximum wave setup. The authors train and test a GP model with 9 different data sets of wave setup measurements. The pieces used by the GP model to build the equations are 6 variables and 6 operators, which are among those used in previous setup formulas. Then, they compare the obtained predictors with other 7 existing formulas for maximum wave setup. They obtain an extremely simple predictor that give wave setup with the same accuracy than the best of the previous existing formulas and a more complex expression that outperforms them.*

*Obtaining more accurate formulas for the maximum wave setup is important to increase our capacity to predict flooding since wave setup can contribute significantly to inundation episodes. This is especially crucial in the present framework of climate change and the urgent need of quantifying its potential effects. Beyond the fact that the two new obtained formulas can be directly used, the article also shows the potential of a new methodology (genetic algorithms) to capture the trends and provide accurate formulas for wave setup. This is very interesting since it can be applied to obtain better empirical formulas as soon as new better-quality data is available. The article is well written and their approach and results are of great interest for the coastal research community.*

*My overall impression is very positive but I still have a few comments and suggestions that might improve the manuscript: there are a few extra analyses that I missed in this study (specific comment 1), some parts of their methodology and results should be described in more detail (specific comment 2) and some parts of the text could be synthesized (specific comment 3). Below there is a longer description of these comments, together with a list of possible technical corrections. Overall, I recommend publication after these minor issues have been considered by the authors.*

We appreciate that you found our manuscript interesting and thank you for the positive analysis. Below we address each specific comment in detail.

***1) Potential extra analysis (from more to less important) that could be added:***

*- To my opinion, it would be extremely interesting to apply the two new formulas together with the other 7 to a new data set not included in the training and test sets. This would allow quantifying if the new formulas maintain their better predictive capacity beyond the data that was used to train them.*

This is a valuable suggestion, thank you. Unfortunately, it is still challenging to find freely available data that include not only wave characteristics and beach morphology but also measured wave setup. Moreover, it was beyond the scope of this work to produce new data. Hopefully, soon more data will become available. As suggested, we now recommend in the Discussion that future research should test the formulas using an entirely new data set not included in the training and test sets. Please see below:

"An avenue for future research includes the validation of the GP predictors (Eqs. (10) and (11)) by applying them to datasets not included in the training. This will further assess the predictive capability of the new formulas and the importance of each term in the equations." (lines 291 - 293 in the new version of the manuscript)

*- If available, it might be enlightening to show beach profiles corresponding to the 9 data sets used in this study. Maybe the sets that show less correlation (Duck 94 and SandyDuck) have a specificity. In particular, I would say that Duck beach has a mixture of fine and coarse sand and it often displays a double slope profile, with a larger slope at the foreshore corresponding to a coarser portion and a smaller slope in the rest of the profile linked to the finer portion.*

Thank you for the suggestion. However, we do not have access to the cross-shore transects but only to the mean foreshore slope. Stockdon et al. (2006)'s paper presents the profile of each beach, but due to Copyright reasons, we are not reproducing it in this paper. Based on the available information, it is not clear what is the specificity that differentiates Duck 94 (and SandyDuck) from the others. A possible explanation is now added to the Discussion section as follows:

"Martins et al. (2022) suggest that it might be difficult to differentiate between swash and wave motions near the shoreline in the field, particularly for steeper foreshores. The considerable influence of the swash circulation within a cusp field during the Duck 94 experiment described by Stockdon et al. (2006) could be an explanation for the lowest correlation of the measured data with the GP predictors results. In essence, measuring and accounting for all the effects and processes that may be important for wave setup remains an arduous challenge." (lines 311 - 315 in the new version of the manuscript)

Added reference: Martins, K., Bertin, X., Mengual, B., Pezerat, M., Lavaud, L., Guérin, T., and Zhang, Y. J.: Wave-induced mean currents and setup over barred and steep sandy beaches, Ocean Modelling, 179, 102 110, https://doi.org/10.1016/j.ocemod.2022.102110, 2022.

*- A variable that is important for nearshore processes that do not appear in the study is wave direction with respect to the shore normal. Do the authors have the wave direction corresponding to the different points in their data set? At least, they could plot the mean direction during the experiments and discuss if the sets that show less/more correlation show a specific direction.*

Unfortunately, we do not have wave direction data. However, thank you for the valuable suggestion. We incorporated this comment in the Discussion section as a suggestion for future works. Please see below:

"More importantly, the novel inclusion of $D_{50}$ as a second-order effect may indicate that we still have very limited information to describe an entire beach. Other examples of second-order effects are not considering the presence of multiple bar systems or even incorporating wave direction." (lines 294 - 296 in the new version of the manuscript)

*- I am curious to know what would happen if Eq. (14) is applied using only the two first terms (so without the term related with grain size). Maybe this is out of the scope of this study but what would happen if this is plotted in Figure 5, too? That is, I wonder how sensitive is equation (14) to the last term? The text related to Figure 5 about the role of D50 (lines 240-244) might be expanded by including such analysis, if the authors find this interesting.*

Thank you for the suggestion. We ran the test, and the results are presented in Figures a and b below. Without the last term, the equation does not represent well the measured data, underpredicting wave setup. Looking at the metrics, the coefficient of determination ($R^2$) essentially indicates that the equation predicts as well as blind guesses made around the average observed data. In this sense, this predictor needs the third term. We did not include this analysis in the paper because removing a term from the machine learning (ML) result changes the entire solution. The GP method does not evaluate the physical meaning of parameters but only how well the model predicts. It is our work to choose the best and physically meaningful equation, as pointed out in the first paragraph of the Discussion section. Please see below:

"Although the main advantage of the GP model is the possibility of fully exploring multiple equation forms from different model parameters trying to find a more accurate variable combination during evolution, the final selection of the proposed solution remains subjective. This last step requires the user to have knowledge of the specific topic, so that the expression chosen is dimensionally and physically correct." (lines 263 - 267 in the new version of the manuscript)

[Figure]

Figure a - Measured versus predicted maximum wave setup (ηM) using the testing data for Eq. (10) (former Eq. (13)) (left panel) and Eq. (11) (former Eq. (14)) (right panel). Different markers/colours refer to different field experiments, as referenced in the legend.

[Figure]

Figure b - General behaviour of the maximum wave setup ($\eta_M$) predictors presented and Eq. (11) (former Eq. (14)) with and without the third term, as a function of deep-water significant wave height ($H_{s0}$), Iribarren number ($\xi_0$) and median sediment diameter ($D_{50}$). $D_{50}$ is represented by its minimum (0.2 mm), mean (0.5 mm) and maximum (2.0 mm) values in the dataset. Data within the measured range are depicted with red/green points. Black/blue points represent an extrapolated range for $H_{s0}$ and $\xi_0$.

**2) The following issues should be clarified in the text:**

*- Section 1: How are surf zones slopes and foreshore slopes defined? (e.g. which water depths?).*

According to Stockdon et al. (2006), surf zone slope is the slope between the shoreline (cross-shore position of $\eta_M$), and the cross-shore location of wave breaking, and foreshore slope is the average foreshore slope with respect to the still water level ± twice the standard deviation of the continuous water level. We agree that the information used in our work should have been better described. We now clarified the foreshore slope in the Data section as follows:

"The dataset contains measurements of: maximum setup ($\eta_M$), foreshore beach slope ($\beta_f$ ) - average foreshore slope with respect to still water level ± twice the standard deviation of the continuous water level, and associated offshore wave characteristics ($H_{s0}$ – significant wave height, and $T_p$ – peak period) from 10 field experiments on sandy beaches resulting in a total of 491 measurements." (lines 111 - 114 in the new version of the manuscript)

*- Line 124: Iribarren number in the paper is computed with the foreshore slope, right? This is not what was used in the previous study where it first appear and thereby in the definition of line 58. Please,*

Thank you for pointing out this confusion. The Irribaren number presented by Holman and Sallenger Jr. (1985) is also computed using the foreshore slope. This has been corrected and the following sentence in the text now becomes clear, as pointed out in the technical corrections (lines 58-60). Please see below:

"Later, Holman and Sallenger Jr (1985) found a more accurate correlation than the one presented by Guza and Thornton (1981) by relating the setup with the surf similarity parameter (Iribarren number:

$\xi_0 = \beta_f /(H_{s0}/L_0)^{0.5}$, where $\beta_f$ is the foreshore slope, $H_{s0}/L_0$ is the wave steepness and $L_0$ the offshore wavelength). However, when isolating low tide data, no significant trend was found with $\xi_0$, indicating the probable setup dependency on the entire surf zone's bathymetry and not only on the foreshore slope." (lines 47 - 51 in the new version of the manuscript)

*- Section 2: The authors should justify the choice of the variables in their GP model. The 6 used variables sound completely reasonable to me given the previous predictors published in the literature but I think the paper would benefit from a sentence arguing this.*

The only freely available dataset containing setup, wave characteristics, and beach morphology parameters was from Stockdon et al. (2006). We included a sentence at the beginning of the Data section clarifying this and also indicated in the second paragraph of the Conclusion the need of additional variables. Please see below:

"To make setup predictions using a data-driven model, it is necessary to have the input and output data to train it. The input data is related to physical processes that induce the output, wave setup. In this work, we used a dataset meeting these requirements, representing a large variety of beach and wave conditions compiled by Stockdon et al. (2006)." (lines 108 - 110 in the new version of the manuscript)

"So far, only a few studies have addressed wave setup predictions, and all past predictors present significant scatter around the data. All predictors share similarities in their structure, possibly indicating that limits in predictability are related to the use of oversimplified variables, $H_{s0}$, $T_p$, $\beta_f$, and $D_{50}$, that do not fully capture the complexity of surf zone processes. The use of additional parameters (e.g., to better describe the surf zone seabed profile and nearshore circulation processes) appears necessary to more accurately describe wave setup in a natural environment." (lines 325 - 329 in the new version of the manuscript)

*Moreover, why don't they also include beta_s (surf zone slope)?*

Unfortunately, we do not have datasets containing surf zone slope and wave setup (as well as other variables). In the conclusion, we have mentioned the necessity of additional parameters and data to improve wave setup predictions (Please see previous answer).

*Related to this, a comment about the introduction. Somewhere at the beginning of page 3, it would be nice to summarize what are the variables that have been used in the existing predictors (that are presented in the following paragraphs). The variables keep popping up each time a new predictor is presented and I think that anticipating potential variables at the beginning of page 3 would improve the text.*

We thank you for the suggestion, but we have decided not to include this list of variables previously in the text. We believe this prevent duplication and it is easier for the reader to learn the meaning of each variable as they advance in the text other than needing to return to the beginning each time a new variable appears.

*- Section 2: Also, the choice of operators in the GP model should be justified. In particular, what does it mean x^x? Why is this operator chosen instead of x^y or x^const?*

Thank you for pointing out this confusion. We agree that x^x was not the best representation for a power function and x^y (with y being a constant) is in fact clearer. In any case, we have decided to

delete it from the table and add it only in the text since it was not used to find the current predictors in the end, only in the search. See the corrected text below:

"We have run the model with different setups such as different mathematical operators (addition, subtraction, multiplication, division, square root, power, log, absolute, inverse, sine, cosine, tangent), population sizes (2,000 - 500,000), generations (20 - 10,000), tournament sizes (10 - 1,000), parsimony coefficients (0.0005 - 0.01) and genetic operations proportions." (lines 175 - 178 in the new version of the manuscript)

*Also, why the parameter values are limited from -5 to +5?*

We chose this range of constants after testing values outside this range that resulted in poor predictions. This is clarified in Table 2 as follows:

"The choice of the values above was driven by extensive testing and sensitivity analyses performed as part of this work."

*- Line 174: Clarify what is a parsimony coefficient and how does it work.*

We have clarified the meaning and the use of the parsimony coefficient in the text as follows:

"A parsimony coefficient is used to penalize long equations, avoiding bloat (longer equations with no significant improvement in fitness). It is used during a tournament to deduct from the fitness result of the longer equation among two competitors that present identical results, the longer one being discarded." (lines 167 - 170 in the new version of the manuscript)

*- Section 3: How the GP model arrives at the factor 1625 in Eq. (14)? What is written in the Methods section is that constant range is from -5 to +5 (e.g. Table 2).*

The original mathematical expression resulting from the GP model was the one below. However, we did simplify it to the one found in the paper (originally Eq. 14, now Eq. 11). We have clarified this information in the text as follows:

"Also, we did simplify the Eq. (11) from the original one, presenting fewer coefficients but with the same output." (lines 211 - 212 in the new version of the manuscript)

$$\eta_M = \frac{H0}{4.08} * \left( \frac{\xi 0}{3.25} + \frac{\frac{\xi 0}{3.25}}{\frac{\xi 0}{3.25} + 0.197} + \frac{\frac{\xi 0}{3.25}}{\frac{D50}{0.002} + \frac{\xi 0}{3.25}} \right)$$

*- Section 3: How robust is the result of the GP model? How the final formulas are chosen? Every time it is run, even with the same input parameter values of Table 2, it provides different formulas like Eq. (13) and (14)? Or it manages to always converge to these two selected ones?*

Every time we run the model with the exact same setup, the result is the same. If the input parameters are changed, the algorithm usually converges to a different solution although at times the same form as Eqs. (10) and (11) (former Eqs. (13) and (14), respectively) can be obtained. We have clarified this information in the footnote of Table 2 (see below). Also, as indicated in the Methodology Section, it is the role of the operator to choose which formula is the most appropriate and this has been done on the basis of physical interpretation and robustness of the predictions beyond the training range.

"The choice of the values above was driven by extensive testing and sensitivity analyses performed as part of this work. The final model setup for each equation varies slightly. Details of both final codes are available and can be accessed through the Code and data availability section."

*- In line 233, the authors write that Duck94 showed less correlation than the rest. However, watching in detail at Fig. 4, my impression is that SandyDuck also stands out, at least in Eq. (13) plot.*

Although some SandyDuck data stand out in part of its data points, as seen in Figure 4, the remaining data is well represented by the new predictors. The same does not happen with Duck 94, since the new predictors underestimate almost all data points. When plotting both datasets only (and the metrics – Figure c), it is easy to visualize these differences. The metrics presented show that the SandyDuck dataset performs better than Duck 94.

[Figure]

Figure c - Measured versus predicted maximum wave setup ($\eta_M$) using the testing data for Eq. (10) (former Eq. (13)) (left panel) and Eq. (11) (former Eq. (14)) (right panel). Different markers/colours refer to different field experiments, as referenced in the legend.

*- Line 254: The authors mention that the best of the previous models (Ji et al., 2018) has the disadvantage of having one coefficient more than Eq. (13). What do they mean?*

Thank you for pointing out this mistake. This statement is not correct. Both equations have the same number of variables and coefficients in the formula. We have deleted this sentence from the text.

*- Line 270-271: Ji et al. (2018) formula also presents a good fit for dissipative and reflective conditions, right? This should be acknowledged.?*

Yes, this is true. We have acknowledged this in the Results section as follows:

"Overall, our ML-driven approach achieved better results with Eq. (11) outperforming all other predictors. Similarly, Eq. (10) exhibits good results, the same ones as Ji et al. (2018)'s equation, which also performs well on dissipative and reflective beach conditions." (lines 247 - 249 in the new version of the manuscript)

**3) The text is in general well written and to the point but, to my opinion, the following parts could be synthesized:**

*- Delete lines 18-24 in the Intro. This contains standard knowledge, included in any book on nearshore processes, it is not needed in research articles, in my opinion.*

We agree and have deleted those lines.

*- Delete or synthesize lines 40-45 in the Intro. Is this adding something, given that most of the formulas below are purely empirical? At least, the formula could be deleted, maintaining only a summary of the main findings of Bowen et al. (1968). Alternatively, when introducing the work of Battjes (1974), the previous result of Bowen et al. (1968) could be acknowledged.*

We agree and have synthesised those lines. One of the main findings of Bowen et al. (1968) was that the theory, based on the concept of radiation stress, underpredicts measured wave setup values. We believe this information is important, therefore we have maintained it. Please see below:

"In one of the first studies about setup, Bowen et al. (1968) conducted a laboratory investigation with monochromatic waves. Their results indicated that the theory, based on the concept of radiation stress, underpredicts measured setup values, especially at the shoreline. The maximum setup ($\eta_M$), time-averaged elevation of the water level at the shoreline, became the focus of subsequent studies." (lines 32 - 35 in the new version of the manuscript)

*- Lines 250-258 could be more to the point. For example, I think it is unnecessary to write all the numbers of Table 3 (they can be seen on the Table).*

We agree with the suggestion and have deleted the metrics presented in the text.

***Technical corrections***

We thank you Dr. Ribas for the careful reading. All the suggestions in technical corrections have been accepted and corrected in the updated version of the paper. The only exception was the one related to figure 5. Please see the comments below:

*Line 15: According to the same authors, as -> As* – Corrected (line 15).
*Line 28: such as -> including [I understand the authors write here two examples of nearshore currents that are especially sensitive to setup, other currents being less sensitive to it.]* – Corrected (line 21).
*Line 29: in the flow circulation and so to sediment exchanges -> in the flow and sediment exchanges*
*Line 34: Delete "using Eq. (2) as the initial point" since many of the references use empirical approaches, right? [in fact, I suggest to delete Eq, 2]* – Removed.
*Line 46: Define in one sentence \eta_M as (= maximum setup, which always occurs at the shoreline). Being the main character of your story, it deserves a careful definition, right?* – Defined: "The maximum setup ($\eta_M$), time-averaged elevation of the water level at the shoreline, became the focus of subsequent studies." (lines 34 - 35)
*Line 57: than only using Hs0 by relating setup and the surf zone similarity parameter -> by relating setup with the surf zone similarity parameter* – Corrected (line 48).
*Line 58: L0 should be defined here instead of after Eq. (6). Also, the authors could already introduce wave steepness here, instead of doing it before Eq. (7).* – Corrected (lines 49 - 50).
*Lines 58-60: Since "foreshore slope" is not mentioned before in the text, I do not understand this sentence.* – Corrected (line 49).
*Line 63: I suggest to write Eq. (6) right after it is mentioned for the first time, so before the sentence that now starts "The equation (Eq. 6)…"* – Corrected (line 55).

*Line 70: was also -> had also been [I would say]* – Corrected (line 61).

*Lines 86-87: simplifications, uncertain or -> as well as simplifications, uncertainty or* – Corrected (lines 77 - 78).

*Lines 113-114: This sentence belongs more to the Introduction, to my opinion.* – Removed.

*Line 129: conditions with -> conditions, with* – Corrected (line 120).

*Table 1: Since every data set occupies two lines in the table, I suggest to add horizontal lines to separate between data sets and increase readability.* – Corrected.

*Figure 2: It should be enlarged significantly to make it readable. Probably it should be a horizontal 1-page figure.* – We've enlarged numbers and letter fonts. The figure size is according to the journal's guidelines.

*Line 170: From the best solution, a new solution is created -> From the best solutions, a new set of solutions is created [What understand is that this step is done after a new generation has been created, in order to create the next generation. Thereby many solutions are created, right? Also, to make a crossover you need to combine at least 2 parent equations…]* – Corrected (line 163).

*Line 181: The range of parsimony coefficients in the text do not include the value finally used (Table 2).* – Corrected (line 177).

*Lines 207-209: Order the citations by year, as in the rest of the article* – Corrected (lines 203 - 204).

*Figures 3-6: They could be slightly enlarged to make them more readable (or maybe only enlarge numbers and letters font).* – We've enlarged numbers and letter fonts. The figure size is according to the journal's guidelines.

*Line 212: predictor which -> predictor, which [I would say that which sentences must be between commas]* – Corrected (line 207).

*Line 213: interpretability is -> interpretability, is [I would say that which sentences must be between commas]* – Corrected (line 208).

*Line 217: This equation is -> It is* – Corrected (line 213).

*Figure 5: My impression is that showing the results of the two equations in the same panel would be more illustrative, since they could be more easily compared.* We have tested it, but when both equations are plotted together, they mostly overlap, presenting a similar behaviour and being difficult to distinguish one from another. The idea of this figure is not to compare the equations but to show that both have coherent behaviour with the real environment, including using data that the model was not trained or tested on (i.e., extrapolated data).

*Line 279: presented in the equations are the same used-> present in the two obtained predictors are the same as those used* – Corrected (line 272).

*Line 284: was also introduced -> also appears* – Corrected (line 277).

*Line 298: The sentence starting with "The novel inclusion…" could start a new paragraph, given that this one and the following sentences cover a different topic, right?* – Corrected (line 294).

*Line 304: Equipments as -> Applied equipments include* – Corrected (line 303).

*Line 330: Currently, innovative data-driven approaches, such as genetic programming -> Innovative data-driven approaches, such as the genetic programming applied in this study* – Corrected (line 330).

*Line 336: We expect that the results of this work will -> The results of this work can* – Corrected (lines 334 - 335).

**Once again, we thank the reviewers very much for the time and careful consideration of our study. These have greatly improved the quality of our manuscript.**

**Kind regards, The Authors.**

References:

Battjes, J. A.: Computation of set-up, longshore currents, run-up and overtopping due to wind-generated waves, Ph.D. thesis, Delft University of Technology, http://resolver.tudelft.nl/uuid:e126e043-a858-4e58-b4c7-8a7bc5be1a44, 1974.

Beard, D. C. and Weyl, P. K.: Influence of texture on porosity and permeability of unconsolidated sand, AAPG bulletin, 57, 349–369, https://doi.org/10.1306/819A4272-16C5-11D7-8645000102C1865D, 1973.

Bowen, A., Inman, D., and Simmons, V.: Wave 'set-down'and set-up, Journal of Geophysical Research, 73, 2569–2577, https://doi.org/10.1029/JB073i008p02569, 1968.

Gallagher, E. L., Elgar, S., and Guza, R.T.: Observations of sand bar evolution on a natural beach, Journal of Geophysical Research, 103, 3203–3215, https://doi.org/10.1029/97JC02765, 1998.

Gomes da Silva, P., Medina, R., González, M., and Garnier, R.: Observations of wave, runup and beach characteristics during the MUSCLE-Beach Experiments, Mendeley Data, V4, doi: 10.17632/6yh2b327gd.4, 2018.

Holland, K., Raubenheimer, B., Guza, R. T., and Holman, R. A.: Runup kinematics on a natural beach, Journal of Geophysical Research: Oceans, 100, 4985–4993, https://doi.org/10.1029/94JC02664, 1995.

Holman, R. A. and Sallenger Jr, A.: Setup and swash on a natural beach, Journal of Geophysical Research: Oceans, 90, 945–953, https://doi.org/10.1029/JC090iC01p00945, 1985.

Ji, C., Zhang, Q., and Wu, Y.: An empirical formula for maximum wave setup based on a coupled wave-current model, Ocean Engineering, 147, 215–226, https://doi.org/10.1016/j.oceaneng.2017.10.021, 2018.

Krumbein, W. C. and Monk, G. D.: Permeability as a function of the size parameters of unconsolidated sand, Transactions of the AIME, 151, 153–163, https://doi.org/10.2118/943153-G, 1943.

Longuet-Higgins, M. S.: Wave set-up, percolation and undertow in the surf zone, Proceedings of the Royal Society of London. A. Mathematical and Physical Sciences, 390, 283–291, https://doi.org/10.1098/rspa.1983.0132, 1983.

Martins, K., Bertin, X., Mengual, B., Pezerat, M., Lavaud, L., Guérin, T., and Zhang, Y. J.: Wave-induced mean currents and setup over barred and steep sandy beaches, Ocean Modelling, 179, 102 110, https://doi.org/10.1016/j.ocemod.2022.102110, 2022.

Mason, C., Sallenger, A. H., Holman, R. A., and Birkemeier, W. A.: Duck 82 – A Coastal storm processes experiment, Coastal Engineering Proceedings, 1, pp. 1913–1928, https://doi.org/10.9753/icce.v19.128, 1984.

Nielsen, P.: Wave setup: A field study, Journal of Geophysical Research: Oceans, 93, 15643–15652, https://doi.org/10.1029/JC093iC12p15643, 1988.

Nielsen, P.: Wave setup and runup: An integrated approach, Coastal Engineering, 13, 1–9, https://doi.org/10.1016/0378-3839(89)90029-X, 1989.

Passarella, M., Goldstein, E. B., De Muro, S., and Coco, G.: The use of genetic programming to develop a predictor of swash excursion on sandy beaches, Natural Hazards and Earth System Sciences, 18, 599–611, https://doi.org/10.5194/nhess-18-599-2018, 2018.

Poate, T. G., McCall, R. T., and Masselink, G.: A new parameterisation for runup on gravel beaches, Coastal Engineering, 117, 176–190, https://doi.org/10.1016/j.coastaleng.2016.08.003, 2016.

Power, H. E., Gharabaghi, B., Bonakdari, H., Robertson, B., Atkinson, A. L., and Baldock, T. E.: Prediction of wave runup on beaches using Gene-Expression Programming and empirical relationships, Coastal Engineering, 144, 47–61, https://doi.org/10.1016/j.coastaleng.2018.10.006, 2019.

Raubenheimer, B. and Guza, R. T.: Observations and predictions of run-up, Journal of Geophysical Research: Oceans, 101, 25575–25587, https://doi.org/10.1029/96JC02432, 1996.

Raubenheimer, B., Guza, R., and Elgar, S.: Field observations of wave-driven setdown and setup, Journal of Geophysical Research: Oceans, 106, 4629–4638, https://doi.org/10.1029/2000JC000572, 2001.

Ruessink, B. G., Kleinhans, M. G., and Van den Beukel, P. G. L.: Observations of swash under highly dissipative conditions, Journal of Geophysical Research: Oceans, 103, 3111–3118, https://doi.org/10.1029/97JC02791, 1998.

Ruggiero, P., Holman, R. A., and Beach, R. A.: Wave run-up on a high-energy dissipative beach, Journal of Geophysical Research: Oceans, 109, https://doi.org/10.1029/2003JC002160, 2004.

Shepherd, R. G.: Correlations of permeability and grain size, Groundwater, 27, 633–638, https://doi.org/10.1111/j.1745-6584.1989.tb00476.x, 1989.

Stauble, D.K. and Cialone, M.A.: Sediment dynamics and profile interactions: Duck94, Coastal Engineering Proceedings, pp. 3921–3934, https://doi.org/10.1061/9780784402429.303, 1996.

Stockdon, H.F. and Holman, R.A.: Observations of wave runup, setup, and swash on natural beaches, U.S. Geological Survey Data Series 602, 2011. [https://pubs.usgs.gov/ds/602/]

Stockdon, H. F., Holman, R. A., Howd, P. A., and Sallenger Jr, A. H.: Empirical parameterization of setup, swash, and runup, Coastal Engineering, 53, 573–588, https://doi.org/10.1016/j.coastaleng.2005.12.005, 2006.

Thornton, E. B., Humiston, R. T., and Birkemeier, W.: Bar/trough generation on a natural beach, Journal of Geophysical Research: Oceans, 101, 12097–12110, https://doi.org/10.1029/96JC00209, 1996.

Tinoco, R., Goldstein, E., and Coco, G.: A data-driven approach to develop physically sound predictors: Application to depth-averaged velocities on flows through submerged arrays of rigid cylinders, Water Resources Research, 51, 1247–1263, https://doi.org/10.1002/2014WR016380, 2015.

Wang, Y., Chen, J., Cai, H., Yu, Q., and Zhou, Z.: Predicting water turbidity in a macro-tidal coastal bay using machine learning approaches, Estuarine, Coastal and Shelf Science, 252, 107276, https://doi.org/10.1016/j.ecss.2021.107276, 2021.

Ward, J. C.: Turbulent flow in porous media, Journal of the Hydraulics Division, 90, 1–12, https://doi.org/10.1061/JYCEAJ.0001096, 1964.

---

## Author Response (AR2)

Dear Dr. Ribas,

**We want to thank you for the valuable review of our manuscript. We agree with the constructive comments and have modified the manuscript to take these on board. Please see below our responses to your comments.**

*The authors have carefully considered and replayed to all the comments and suggestions of the two reviewers, including all those that were feasible in the text. To my opinion, the article can already be published mostly as it is.*

We thank you for the positive analysis of our manuscript.

*I only recommend the authors to doube-check the following two technical corrections:*

**1) Revise one of the new sentences, which is somewhat confusing to me:**

*"It is used during a tournament to deduct from the fitness result of the longer equation among two competitors that present identical results, the longer one being discarded." (lines 169-170). Maybe you can simply write: "It is used during a tournament to discard from the fitness results the longer equation among two competitors that present identical results."*

Thank you for pointing out this confusion. We agreed with your suggestion and have rewritten the sentence as recommended (lines 168-169).

**2) Revise the sentence:** *"Correlation and agreement results showed by O'Grady et al. (2019)'s equation are also close to Eqs. (10) and (4) (Ji et al., 2018)." (lines 250-251). What does (4) stands for? Do you mean eq. (4) of Ji et al. (2018)? You did not mention this eq. number in all the previous citations to Ji et al. (2018)... Please, clarify.*

Yes, (4) stands for eq. (4) of Ji et al. (2018). Thank you for pointing this out. We agree it is not clear. At the beginning of this paragraph, we refer to Table 3 and follow the same pattern for all other equations except the one you mentioned. We have now clarified it in the text as follows.

"Correlation and agreement results showed by O'Grady et al. (2019)'s equation are also close to Ji et al. (2018)'s and Eq. (10) in this work." (lines 249-250)

**Once again, we thank the reviewers very much for the time and careful consideration of our study. These have greatly improved the quality of our manuscript.**

**Kind regards, The Authors.**